# A Review of Recent Advances in Computer-Aided Detection Methods Using Hyperspectral Imaging Engineering to Detect Skin Cancer

**DOI:** 10.3390/cancers15235634

**Published:** 2023-11-29

**Authors:** Hung-Yi Huang, Yu-Ping Hsiao, Riya Karmakar, Arvind Mukundan, Pramod Chaudhary, Shang-Chin Hsieh, Hsiang-Chen Wang

**Affiliations:** 1Department of Dermatology, Ditmanson Medical Foundation Chiayi Christian Hospital, Chia Yi City 60002, Taiwan; huanghungyi1@gmail.com; 2Department of Dermatology, Chung Shan Medical University Hospital, No.110, Sec. 1, Jianguo N. Rd., South District, Taichung City 40201, Taiwan; missyuping@gmail.com; 3Institute of Medicine, School of Medicine, Chung Shan Medical University, No.110, Sec. 1, Jianguo N. Rd., South District, Taichung City 40201, Taiwan; 4Department of Mechanical Engineering, National Chung Cheng University, 168, University Rd., Min Hsiung, Chia Yi City 62102, Taiwan; karmakarriya345@gmail.com (R.K.); d09420003@ccu.edu.tw (A.M.); 5Department of Aeronautical Engineering, Vel Tech Rangarajan Dr. Sagunthala R&D Institute of Science and Technology, Avadi, Chennai 600 062, India; chaudharyranch@gmail.com; 6Department of Plastic Surgery, Kaohsiung Armed Forces General Hospital, 2, Zhongzheng 1st. Rd., Lingya District, Kaohsiung 80284, Taiwan; 7Department of Medical Research, Dalin Tzu Chi General Hospital, No. 2, Min-Sheng Rd., Dalin Town, Chia Yi City 62247, Taiwan; 8Technology Development, Hitspectra Intelligent Technology Co., Ltd., Kaohsiung 80661, Taiwan

**Keywords:** skin cancer, hyperspectral imaging, meta-analysis, melanoma

## Abstract

**Simple Summary:**

Recent advancements in this field have shown continuous improvement, with CAD algorithms enhancing diagnosis accuracy. The study systematically analyzes various aspects of HSI research for skin cancer detection, emphasizing global efforts and the need for more collaboration and data sharing. The results suggest promising advancements in skin cancer identification through HSI, offering potential for early diagnosis and tailored treatments. Integrating CAD algorithms improves diagnostic accuracy. Despite challenges such as limited patient involvement and data availability, HSI offers non-invasive, precise lesion characterization, reducing false positives and negatives. It holds the potential to improve early detection and treatment planning. With ongoing research and collaboration, HSI holds the potential to save lives and improve patient care by facilitating early diagnosis and personalized treatment.

**Abstract:**

Skin cancer, a malignant neoplasm originating from skin cell types including keratinocytes, melanocytes, and sweat glands, comprises three primary forms: basal cell carcinoma (BCC), squamous cell carcinoma (SCC), and malignant melanoma (MM). BCC and SCC, while constituting the most prevalent categories of skin cancer, are generally considered less aggressive compared to MM. Notably, MM possesses a greater capacity for invasiveness, enabling infiltration into adjacent tissues and dissemination via both the circulatory and lymphatic systems. Risk factors associated with skin cancer encompass ultraviolet (UV) radiation exposure, fair skin complexion, a history of sunburn incidents, genetic predisposition, immunosuppressive conditions, and exposure to environmental carcinogens. Early detection of skin cancer is of paramount importance to optimize treatment outcomes and preclude the progression of disease, either locally or to distant sites. In pursuit of this objective, numerous computer-aided diagnosis (CAD) systems have been developed. Hyperspectral imaging (HSI), distinguished by its capacity to capture information spanning the electromagnetic spectrum, surpasses conventional RGB imaging, which relies solely on three color channels. Consequently, this study offers a comprehensive exploration of recent CAD investigations pertaining to skin cancer detection and diagnosis utilizing HSI, emphasizing diagnostic performance parameters such as sensitivity and specificity.

## 1. Introduction

Skin cancer is mainly encountered in people with a lighter skin complexion [1]. It can most often be found in countries like the United States of America, Germany, China, and France [2]. Skin cancer currently represents one-third of all cancer diagnoses worldwide, and the number of cases has been continuously increasing in recent years [3]. Skin cancer can be classified as non-melanoma skin cancer (NMSC) or melanoma [4]. In 2018, non-small cell lung cancer (NMSC) was the fifth most common form of cancer worldwide (excluding basal-cell carcinomas, or BCCs), accounting for over one million different detections and approximately sixty-five thousand deaths, while malignancy was the current century’s most common form of cancer, accounting for nearly 300,000 new cases and 60,000 deaths [5,6,7,8,9]. The prevalence of the two types of non-melanoma and melanoma cancers of the skin has exhibited an upward trend in recent decades [10,11,12]. Presently, the annual incidence of non-melanoma skin cancers (NMSC) ranges from 2 to 3 million cases worldwide, whereas the occurrence of melanoma skin cancers amounts to approximately 132,000 cases globally. [13]. The estimated number of new cases of skin cancers (excluding BCC and SCC) in the US in 2022 is 108,480, with 62,820 in males and 45,660 in females [14,15,16,17]. The total number of melanoma skin cancers is 99,780, with 57,180 in males and 42,600 in females [18]. There are 8700 cases of other non-epithelial skin cancer, with 5640 in males and 3060 in females [19]. Among these, the estimated deaths of skin cancers in the US in 2022 were 11,990, with 8060 males and 3930 females [20,21,22]. In melanoma skin cancer, the estimated death cases are 7650: 5080 in males and 2570 in females [23]. Out of the 8700 other non-epithelial skin cases, the estimated number of mortality cases is 4340: 2980 for males and 1360 for females [24]. In a study, the analysis examines the ten-year rate of survival for melanoma individuals in Japan between 1987 and 2001 [25]. The data indicates that the survival rate among female patients was comparatively greater than that among male patients. Specifically, the 140-month survival rate was found to be 70.6% for females, while it stood at 60% for males [26]. Carcinoma was the leading cause of mortality among individuals diagnosed with skin cancer and blackfoot disease [27]. After the commencement of blackfoot illness, the five-year survival rate was 76.3%, the 10-year survival rate was 63.3%, and the 15-year survival rate was 52.2% [28]. Sixteen years after the first symptoms of the illness appeared, the survival rate dropped to 50 percent [29,30,31].

Computer-aided diagnosis (CAD) is good for cancer detection because it uses artificial intelligence, machine learning models, algorithms, and data acquisition from automated or computerized tools [32,33,34,35]. Zhiying et al. conducted a study in which they used an advanced method of image segmentation that was based on the convolutional neural network (CNN) specifically developed by satin bowerbird optimization (SBO). The study’s primary objective was to reduce image noise in order to achieve higher levels of productivity, as shown by the confusion matrix [36]. In another study by Jaleel et al., imaging techniques and artificial intelligence using artificial neural network (ANN) machine learning technology were used for skin diagnosis instead of going to the hospital [37,38,39]. Biosensors are devices that are designed to detect a specific biological analyte by essentially converting a biological entity into an electrical signal that can be detected and analyzed [40,41,42]. The technology of biosensors has the ability to enable rapid and precise detection, dependable imaging of cancerous cells, and management of cancer spread and angiogenesis [43]. Research conducted by Keshvarz et al. uses water-based tetrahertz metamaterial as a biosensor for the early detection of skin by analyzing image features and characteristics [44]. In another study, Bohunickey et al. used Indium Gallium Arsenide (InGaAs) as a biosensor to analyze pigmented skin lesions within specified wavelength ranges from 414 nm to 995 nm [45]. Nowadays, CAD and biosensors are usually not utilized for skin cancer diagnosis because the biosensor parameter will change according to pressure [46,47,48] and temperature, and it will sometimes give wrong information about images, which is unsuitable for the early detection of cancer [49]. CAD models are also not effective when the user commands the wrong input during data acquisition, and they will not work effectively [50,51,52,53,54].

One of the non-invasive optical imaging systems that can overcome all the aforementioned challenges and complications is HSI [55]. HSI is capable of combining digital imagery with techniques of spectroscopy, which provides enhanced spectral qualities of a recorded picture both within the visible range of the electromagnetic spectrum as well as beyond it [56,57]. In a hyperspectral image, each pixel at each wavelength is analyzed, resulting in a so-called spectral signature [58]. The spectral signature stores all of the spatial data that correspond to a certain substance or picture and its location in space [59]. It has been shown that quantifiable data on tissue biology may be obtained via spectral signature analysis [60]. The HSI technique can overcome the drawbacks of CAD and biosensors as it will analyze each spectral wavelength and data from the signature spectrum with deep penetration of the materials [61,62]. Hyperspectral imaging (HSI) techniques are applied in various fields, including aerospace [63], food technology [64], agriculture [65], medical field [66,67], astronomy [68], skin cancer [69,70,71], breast cancer [72], remote sensing [73], satellite imaging [74], seed viability study [75], biotechnology [76], biosensor [77], environmental monitoring [78,79], counterfeit detection [80,81,82,83], pharmaceuticals [84], medical diagnose [85,86], forensic science [87], thin films [88], oil and gas [89], microbiology [90], chemical industry [91], esophagus cancer [92], spectrum analysis [93], brain tumor [94], nursing [95], physical therapy [96], and surgery [97].

This review assesses the diagnostic efficacy of CAD algorithms employed in the detection and diagnosis of skin cancer. It evaluates the diagnostic outcomes in terms of specificity and sensitivity, provides a concise overview of the studies, and offers recommendations based on the meta-analysis of various CAD approaches utilized. Following the Section 1, the article is dived into four more sections, including methodology, which gives an overview of the exclusion and inclusion criteria with a quality analysis of the article. The Section 4 presents a comprehensive summary of the results obtained from the literature review, including a description of the clinical features discovered and a succinct explanation of each research study. The Section 5 provides the forest plot and Deek’s funnel plot, while the Section 6 provides a brief overview of the proposed review approach and the achievements of current and previous work.

## 2. Related Studies

There are various traditional methods of skin cancer detection, and early identification is the key to better and more effective treatment of the skin lesions [98]. The knowledge of dermatologists and the results of pathological examinations of biopsy specimens are often relied upon to diagnose skin cancer [99]. The standard imaging methods, such as multispectral imaging (MSI), are used in the morphological processing algorithms that underpin the diagnostic assistance system [100]. In the industry of dermatology, one of the basic guidelines for pigmented skin lesion diagnosis is the ABCD rule [101]. Many characteristics of skin lesions are represented by their corresponding letters in the ABCD rule, and these characteristics include asymmetry of the mole, border irregularity, color uniformity, diameter, and evolving size, shape, or color rule [102]. After this observation, a biopsy is prepared when a dermatologist suspects that the skin lesions are infected [103]. After that, a pathological examination of the material is carried out so that a definite diagnosis may be determined [104,105,106]. A number of methods, depending on image data and techniques, integrate the ABCD principle to aid doctors in their regular diagnostic practice for evaluating and classifying pigmented skin lesions (PSL) [107,108,109]. When applying the ABCD rule to diagnose a skin lesion, a score is assigned for each of the four features of the ABCD rule and combined into a total score [110,111]. The total score determines the level of malignancy of the sample taken, where a higher score means a greater level of malignancy [112]. In clinical experiments, the reported sensitivity and specificity of the ABCD rule are in the ranges of 74–91.6% and 45–67%, respectively [113]. Different types of skin cancer, including BCC, SCC, SK, and non-epithelial skin cancer, are shown in Figure 1 [114].

Due to the fact that traditional technology does not place an emphasis on spatial and spectral information, a normal eye or smart phone is unable to identify melanoma and BCC in the early stages of skin cancer [115,116,117,118]. Over the course of the last several years, scientists from a wide variety of disciplines have collaborated on the expansion and development of novel dermoscopic technologies for the early diagnosis of skin cancer, as well as the formulation of diagnostic criteria and computer algorithms [119,120,121]. For example, the ABCD rule has been extended to ABCDE, where the E represents the evolution of the skin lesion over time [122]. With the advancement of machine learning and computerized algorithms, several research groups have been concentrating on developing automated and semi-automated computational methods for detecting and classifying skin lesions [123,124,125]. In addition, researchers focused on conventional RGB (red, green, blue) imaging techniques and dermoscopic imaging techniques and found out the difference between conventional and dermoscopic imaging techniques. Conventional imaging deals with visual inspection, observation, and changes in shape, size, and color, whereas dermoscopy imaging techniques deal with computerized algorithms and tools and easily differentiate skin melanoma [99,126,127].

Our investigation synthesizes a multitude of recent CAD studies, providing a rigorous assessment of sensitivity and specificity using forest plots and Deek’s funnel plots. Notably, the emphasis is on the significance of these findings, highlighting the specific studies that contribute significantly to the meta-analysis and elucidating how different CAD techniques and geographic factors influence skin cancer detection outcomes. This research not only offers insights into the state-of-the-art in skin cancer detection but also underscores the potential of HSI as a transformative technology in the early diagnosis of this critical medical condition.

## 3. Methodology

This research aims to give insight into novel aggressive HSI technology for the detection of skin cancer in the hopes of elucidating what has been accomplished so far and gaining an understanding of the most significant obstacles that remain. On the basis of this, the accompanying criteria for inclusion and exclusion were developed in order to only include current research papers that focused on identifying skin cancer using HSI. The objective behind this screening was to only include papers that were extremely relevant (Appendix A).

### 3.1. Study Selection Criteria

The aim of this research is to conduct an analysis of the progress made in the detection and diagnosis of skin cancer using HSI technology. The objective of this study is to provide a thorough assessment of the benefits and drawbacks related to the use of this technology within the domain of skin cancer diagnosis. This review focuses on research that conforms to the defined inclusion criteria. The selection criteria for research are those that provide explicit numerical outcomes, such as dataset size, recall rate, precision, accuracy, wavelength, and area under curve (AUC). The research should prioritize the investigation of the use of HSI in the identification and diagnosis of skin cancer. Furthermore, it is essential that the studies under consideration have been published within the last ten years and are research papers published in journals that are indexed in the SCI and Scopus databases. In addition, it is important to choose journals having an impact factor of at least 3, an H-index of 50 or higher, and belonging to the top quartile (Q1) of their respective disciplines. The chosen studies were subject to certain inclusion criteria, which were as follows: The inclusion criteria for this study are as follows: (5) studies using either a prospective or retrospective design; (6) studies produced in the English language; and (7) research articles that explicitly concentrate on the detection of skin cancer using hyperspectral imaging (HSI). In addition to taking into account the specified inclusion criteria, this review also eliminates studies that fulfill the predetermined exclusion criteria. The research’s exclusion criteria include the following: (1) studies without adequate data; (2) studies falling under the narrative, systematic review, and meta-analysis categories; (3) comments, proceedings, or study protocols; and (4) conference papers. The QUADAS-2 tool is introduced by the author as a means of assessing the methodological quality of the papers being examined [128,129,130,131]. The tool has four domains, namely, “patient selection”, “index test”, “reference standard”, and “flow and timing”. Additionally, the first three domains are accompanied by an evaluation of their “applicability”. The assessment included the categorization of each component into one of three categories: high risk, low risk, or uncertain risk of bias [132]. The clinical significance of a diagnostic test is indispensable, as it hinges on the robust evaluation of its sensitivity and specificity, metrics that gauge its ability to accurately identify true positive instances and true negative cases, respectively. Sensitivity underscores the test’s capacity to effectively identify individuals with the ailment, while specificity addresses its aptitude to correctly rule out the disease in those without it. A balanced evaluation considers both aspects, ensuring a harmonious equilibrium between identifying those in need of treatment and minimizing unwarranted actions. Moreover, these values are clinician-friendly and integral to the decision-making process, with high sensitivity serving to prevent the oversight of individuals with the disease and high specificity reducing false alarms. Sensitivity and specificity also facilitate comparative analyses, enabling a direct assessment of a novel diagnostic tool against established benchmarks, a critical feature for doctors and researchers. Furthermore, the reporting of sensitivity and specificity data aligns with the requirements of clinical recommendations and regulatory authorities, facilitating evidence-based practice and informed decision-making.

### 3.2. QUADAS-2 Results

Table 1 displays the QUADAS-2 results of the 10 studies that were used in this literature review (Appendix A). The research encompasses the analysis of aspects pertaining to the significance and impact of bias on their findings [133,134,135]. Each study was evaluated based on the following criteria: patient selection, index test, reference standard, flow and timing, and risk of bias (for further details about the inclusion, exclusion criteria and the quality analysis please refer Appendix A). Furthermore, the assessment included considerations pertaining to the suitability of the study population, the performance of the diagnostic test under investigation, and the established benchmark against which it was compared.

## 4. Results

This section provides a detailed overview of the findings derived from the literature evaluation, including a delineation of the identified clinical characteristics and a concise elucidation of each study. This section presents the quantitative findings obtained from each research study. This section presents an examination of the outcomes in relation to recall, precision, and confidence interval (CI). This part establishes a relationship between the tabulated data and utilizes the summary of these findings to provide a precise and accurate assessment of skin cancer.

### 4.1. Studies under Clinical Feature Observation

The objective of this study is to do an analysis of prior research that examines the efficacy of several CAD methods in the specialized field of skin cancer detection. This review provides a concise overview of the included study, with an emphasis on elucidating the objectives, the used CAD approaches, and the resultant discoveries. In addition, subgrouping and meta-analysis approaches were used to ascertain the accuracy, precision, and AUC values associated with the detection and classification of skin cancer lesions. This study primarily focused on the assessment of PSLs and N-PSLs, as outlined in the corresponding literature. The factors under consideration were subjected to testing and comparison utilizing several CAD approaches, as documented in the available literature.

Leon et al. used HSI as a diagnostic modality for the timely identification of skin cancer, with the objective of enhancing therapeutic results and promoting technical advancements in the field of skin cancer detection. The research included a cohort of 61 individuals from a medical facility located in Germany. These participants underwent the acquisition of 76 hyperspectral pictures of photoluminescent substances across the spectral range spanning from 450 nm to 950 nm. The dataset was then used for diagnostic purposes, applying the support vector machine (SVM) technique. The investigation yielded sensitivity and specificity values of 87.5% and 100%, respectively, suggesting a notable degree of precision in the diagnostic process. The study produced an area under the receiver operating characteristic curve (AUC) value of 0.89, suggesting positive numerical results. Lindholm et al. conducted an independent study with the objective of using three-dimensional hyperspectral imaging (3D HSI) technology to detect occurrences of skin cancer on complex skin surfaces. The CNN algorithm was used to detect skin lesions by using HSI on complex skin surfaces. The diagnostic procedure was carried out using a wavelength range spanning from 477 nm to 891 nm. The selection process included the careful selection of 42 skin lesions from a cohort of 33 individuals. The findings of this study demonstrated sensitivity and specificity rates of 87% and 93%, respectively. The aforementioned values are clearly delineated and provide a dependable evaluation of the diagnostic precision.

Christensen et al. used a novel HSI apparatus in their research to detect cutaneous melanoma and PSLs, aiming to improve the effectiveness of early-stage cancer detection in clinical settings. The data collection process for this study included extracting information from a sample size of 186 participants. The research yielded a total of 202 skin lesions, including both primary skin lesions and non-primary skin lesions, which exhibited notable impacts. The use of the DI method in this study has shown significant results, as seen by the notable improvements in sensitivity and specificity values, which reached 96.7% and 42.1%, respectively. The aforementioned results were obtained by diagnostic tests carried out throughout the wavelength range of 400 nm to 800 nm. The study yielded a confidence interval of 95% and an AUC value of 0.800. Hosking et al. undertook research examining the use of HSI in the context of automated digital dermoscopy screening for melanoma. The dataset underwent training and classification using four distinct methodologies, including SVM, k-Nearest Neighbors (k-NN), multilayer perceptron (MLP), and random forest (RF). The study used a sample size of 100 skin lesions procured from a cohort of 91 persons with the objective of conducting a screening for the diagnosis of melanoma. The results of the study demonstrated that the use of HSI in the classification of skin cancer exhibited a sensitivity rate of 100% and a specificity rate of 36%. Nevertheless, it is crucial to acknowledge that the study’s level of accuracy was very low.

In a study done by Pozhar et al., many CAD methods were used. One of the algorithms, known as DI, used the k-NN technique to accomplish the goal of producing precise and extremely accurate outcomes. The use of acousto-optical HSI has shown significant benefits in the detection and diagnosis of skin cancer, especially in its early stages. The dataset was obtained from a sample size of 91 people with the aim of enhancing the identification of skin cancer. The photos were acquired via the SKL method within the wavelength range of 450 nm to 750 nm. The results of this study indicate a significant degree of performance, as shown by an accuracy rate of 78%, a sensitivity rate of 84%, and a specificity rate of 87%. Pardo et al. used HSI as a means to detect melanoma skin cancer by analyzing the spectral attributes of hyperspectral pictures. The dataset included 116 individuals, including a total of 124 skin lesions. The spectrum analysis was performed within the wavelength range of 600 nm to 900 nm using a CNN technique for the purpose of identification. The skin lesions were classified based on their categorization as either melanoma or BCC. The study produced a summary of the following findings: a sensitivity rate of 96.8%, a specificity rate of 95.7%, and an accuracy rate of 96%. The results of this study indicate a significant degree of diagnostic proficiency in the detection of skin cancer.

The research undertaken by Vinokurov et al. focused on the classification of skin disorders using spectral imaging data analysis. The researchers used several CAD techniques to effectively discriminate between distinct categories of skin diseases. The study used a neural network classifier to identify skin problems using HSI technology. The researchers performed a comparative examination of melanoma, BCC, SCC, and SK by using skin disease samples for experimentation within the wavelength spectrum ranging from 450 nm to 950 nm. The findings were derived from a sample size of 648 skin lesions, with a specific emphasis on dermatological conditions, with the aim of differentiating between different forms of skin malignancies. The k-NN approach was used by the authors to calculate the statistical measures of sensitivity (79%) and specificity (95%). Rasanen et al. used HSI to detect unique spectral fluctuations and effectively discriminate between malignant melanoma and pigmented BCC by using a CNN algorithm for the categorization of skin cancer within the wavelength range spanning from 500 nm to 850 nm. The dataset used in this research was acquired from a collective of 26 pigmented lesions, procured from a cohort of 24 people. The results of this investigation indicate a sensitivity of 99.6%, a specificity of 98.6%, and a 95% confidence interval. These values are considered favorable for assessing the quality of this study. Zherdeva et al. undertook research in which they used ex-vivo 45 hyperspectral images with the aim of distinguishing skin cancer by the utilization of CAD techniques. The researchers used the DI, k-NN, SVM, and CNN algorithms for the purpose of analyzing the collected hyperspectral pictures within the wavelength range spanning from 450 nm to 750 nm. The conducted research produced quantitative results, indicating a sensitivity rate of 84% and a specificity rate of 87%. The aforementioned results are deemed exceptional in relation to our study on the use of HSI for the detection of skin cancer. In the investigation conducted by T. Nagaoka et al., a hyper-spectral imager was used in conjunction with an imaging fiberscope to discern and classify different types of skin cancer, specifically discriminating between melanoma and non-epithelial skin cancer. The author used several CAD methods, namely, k-NN, CNN, and SVM, for the purpose of diagnosing hyperspectral images within the wavelength range spanning from 450 nm to 1000 nm. The findings of this study exhibited a sensitivity of 100% and a specificity of 94.4%, suggesting a considerable degree of accuracy and precision. Consequently, these results are deemed appropriate for incorporation into this research endeavor. Huang et al. used the ISIC dataset to detect and classify skin cancer on the basis of BCC, SCC, and SK using a novel snap-shot hyperspectral conversion algorithm that has the ability to convert any RGB image into a HSI image [146]. The recall rates of the RGB and HSI models were 0.722 and 0.794, respectively, thereby indicating an overall increase of 7.5% when using the HSI model.

Table 2 displays the clinical characteristics observed in the clinical trials that were included in the analysis. In the classification of research pertaining to the use of CAD algorithms for the aim of skin cancer diagnosis, two predominant approaches are often utilized: image analysis and patient analysis. This investigation included a range of data collection methods and image processing techniques. All 10 studies included in the analysis specifically reported the number of skin lesions used in their respective research investigations. Additionally, these studies provided comprehensive information on the methodology employed for data collection from hyperspectral images. A cohort of 602 individuals or specimens yielded a total of 1354 skin lesions. The selected papers mostly prioritized the investigation of wavelengths, including visible light and near-infrared light. Five studies were conducted using data from Western countries, while an additional five studies analyzed data from Asian locations. The research employs many CAD algorithms, such as CNN, SVM, DI, SKL, and k-NN. The results of this research were classified into many metrics, including sensitivity, specificity, accuracy, and AUC, which exhibited exceptional numerical outcomes in the detection of skin cancer. The previously stated study categorized the severity of skin cancer lesions, which include BCC, SCC, SK, and melanoma, over a spectrum from moderate to severe.

### 4.2. Meta-Analysis of the Studies

Table 3 presents the results of the meta-analysis and subgroup analysis conducted to examine the diagnosis of skin cancer. In the present study, a total of 10 studies were included. The findings from these studies were synthesized and yielded quantitative outcomes, indicating that the mean values for the number of patients, number of images, recall rate, and specificity rate were 86, approximately 150, 91.89%, and 84.28%, respectively. The categorization of research according to nationality has shown its efficacy in elucidating the distinctions between Asian and Western people. In particular, research conducted on the Asian population revealed a larger sample size of patients and images, along with higher levels of sensitivity (92.12%) and specificity (85.42%) in comparison to studies conducted on the Western population. The latter group exhibited a smaller sample size of patients and images, as well as lower levels of sensitivity (76.35%) and specificity (82.92%). The findings of the research indicate that Asian studies exhibit higher levels of diagnostic sensitivity and specificity when compared to Western studies. The study done by Pardo et al. used a sample size of 124 skin lesions, yielding a sensitivity rate of 96.8% and a specificity rate of 95.7%. Based on the aforementioned meta-analysis, it can be deduced that an augmented volume of data used for training in CAD algorithms resulted in improved diagnostic efficacy. In recent times, there has been a wide-ranging use of CAD algorithms in the realm of skin cancer detection. The results of the meta-analysis suggest that both CNN and SVM have similar levels of sensitivity and specificity, thereby highlighting the efficacy of both CAD algorithms in identifying skin cancer. The sensitivity rate of the CNN algorithm used in skin cancer detection is 91.9%, indicating its ability to correctly identify positive cases. Additionally, the specificity rate is 94.35%, reflecting its capacity to accurately identify negative cases. On the other hand, the SVM approach has a sensitivity rate of 94.56% and a specificity rate of 78.1%. The use of the SKL and DI AI models was infrequent, with the SKL model demonstrating a sensitivity of 84% and a specificity of 87%, while the DI model exhibited a sensitivity of 90.35% and a specificity of 64.55%. SVM and CNN are artificial intelligence techniques that are often used in the domain of skin cancer diagnostics. Nevertheless, the aforementioned techniques have shown divergent numerical values and performance results, resulting in their comparatively worse efficacy when contrasted with the CNN algorithm. The performance of the SVM approach has a high degree of comparability to that of the CNN algorithm, rendering it equally appropriate for the purpose of diagnosing skin cancer. The performance of CNN and SVM exhibited superiority in comparison to other deep learning methods, such as SKL. Hence, drawing from the data obtained via this meta-analysis, it can be deduced that CNN and SVM exhibited greater effectiveness in the detection of skin cancer.

The meta-analysis indicates that the effectiveness of skin cancer diagnosis fluctuates on an annual basis due to developments in HSI technology across the specified time frame. The degrees of sensitivity and specificity have shown fluctuations across time, spanning from earlier to more recent times. The pre-2018 study exhibited sensitivity and specificity rates of 94.25% and 93.85%, respectively, indicating a higher level of performance in comparison to the latest numerical results in the identification of skin cancer after 2018. The sensitivity and specificity values observed in trials performed from 2019 to 2020 demonstrate a decline, with corresponding percentages of 90.5% and 74.33%, respectively. These values show a strong correlation with prior research outcomes. Between the years 2021 and 2022, a considerable cohort of researchers focused their efforts on the advancement of innovative, non-invasive HSI technology. The primary objective of this research endeavor was to facilitate the identification of skin lesions in the diagnostic process of skin cancer. The quantitative results of these endeavors show a marginal enhancement in comparison to the previous temporal interval, spanning from 2019 AD to 2020 AD. The sensitivity and specificity of the studies carried out over the period from 2021 to 2022 showed a respective rise of 0.5% and 8% in their values. The sensitivity and specificity values recorded for the period from 2021 AD to 2022 AD were 90.57% and 82.17%, respectively. Hence, undertaking a progressive study on HSI technology has the potential to offer advantages in augmenting its overall efficacy in the identification and diagnosis of skin cancer.

There is a vast range of skin cancer types that are prominent on a global scale. The risk of developing skin cancer increases as the severity of the problem advances. Different forms of skin cancer, such as BCC, SCC, AK, and melanoma, were classified according to patient count and quantitative data in order to assess their level of severity. The present research investigated a cohort of 276 participants, including 81 persons diagnosed with BCC skin cancer, 91 individuals diagnosed with AK skin cancer, and 104 individuals diagnosed with melanoma. The results of this research suggest an increasing trend in the prevalence of melanoma, a very vulnerable form of skin cancer, among other kinds of skin cancer. The sensitivity and specificity values for BCC are 94.43% and 77.9%, respectively. The study determined that the SCC and AK exhibited sensitivity rates of 90.1% and 92.65%, and specificity rates of 84% and 87%, respectively. Recent studies have shown an increasing incidence of melanoma. A systematic review and meta-analysis were undertaken to assess the diagnostic precision of melanoma, resulting in sensitivity and specificity estimates of 98.62% and 73.36%, respectively. Hence, it may be deduced that there is an upward trajectory in the prevalence of melanoma, a kind of skin malignancy. The quick diagnosis seen in this context may be ascribed to the developments in identification technology and the increased performance characteristics. The results of the meta-analysis suggest that a significant proportion of skin lesions were detected inside the visible near-infrared spectrum, rather than within the visible wavelength range. The sensitivity and specificity values for the spectrum in the visible near-infrared band were 94.567% and 77.3%, respectively. In contrast, the visible wavelength exhibited a sensitivity of 80.4% and a specificity of 92.66%. Hence, it can be inferred that the use of HSI in studies related to the detection of skin cancer may provide more advantages in comparison to the traditional methodology. Furthermore, it is anticipated that there will be improvements in the performance characteristics, such as accuracy, sensitivity, specificity, and AUC.

## 5. Discussion

The graphical representation of the quantitative findings pertaining to skin cancer detection, derived from a meta-analysis of skin cancer studies, was achieved by the use of the forest plot and Deek’s funnel plot. The forest plots were used to analyze the sensitivity and specificity of each CAD approach, taking into account factors such as nationality, kind of skin cancer, band area, year of publication, and the individual studies as shown in Figure 2 (see Appendix A for the Forest plot of sensitivity and specificity based on CAD method, based on Nationality, based on Year of Publication, based on Band Region and based on Type of skin cancers respectively). These analyses were conducted at a 95% confidence level (please refer to Appendix A for the anova analysis of forest plot) [147]. A forest plot is a visual depiction of the findings derived from a meta-analysis, as shown by scholarly sources (refer to Appendix A for the sensitivity and specificity for forest plot cad method, nationality, year of publication, type of skin cancer and band region respectively) [148,149,150]. This study presents the outcomes of research conducted using a 95% confidence interval, including both positive and negative error values [151]. A wider range of confidence intervals is associated with less precise findings, whereas a narrower range of confidence intervals indicates more precision in the obtained results [152]. The dashed line seen in the forest plot symbolizes the threshold for inaction. In the context of the specificity forest plot, the studies conducted by Zherdeva et al. and Pozhar et al. align with the line of no action, indicating that these particular studies have less significance for the pooled meta-analysis. This implies that the *p*-values of the aforementioned studies exceed a confidence interval of 0.005. Specifically, the investigations conducted by Leon et al., Lindholm et al., Pardo et al., Vinokuro et al., Rasanen et al., and T. Nagaoka et al. fall on the positive side of the line of no action, indicating that these studies were statistically significant for the purpose of the meta-analysis. The investigations conducted by Christens et al. and Hosking et al. are positioned on the left side of the line of no action, indicating that these studies do not provide statistically significant results for inclusion in a pooled meta-analysis. The diamond shape is used to represent the magnitude of separate research weights, as well as the precision of their respective findings, as shown by the range of the confidence interval. In the context of the meta-analysis, it can be seen that the studies conducted by Christens et al. and Pardo et al. exhibit lower levels of significance. Conversely, the studies conducted by Hosking et al., Rasanen et al., and T. Nagaoka et al. have higher levels of significance. The investigations conducted by Lindholm et al., Pozhar et al., Vinokuro et al., and Zherdeva et al. are not considered substantial for inclusion in a pooled meta-analysis.

The research concluded by presenting Deek’s funnel plots, which were categorized based on many factors, including the CAD technique, country, skin cancer kind, year of publication, and band region as shown in Figure 3 (see Appendix A for the Deek’s funnel plot based on year of publication, type of skin cancer and wavelength band region respectively). These plots were constructed with a confidence level of 95% for each study included in the analysis. The funnel plot developed by Deek incorporates the odds diagnostic ratio on the *x*-axis and the proportion of the square root of each sample size on the *y*-axis [153,154,155]. The funnel plots developed by Deek provide a comprehensive representation of the regression line and confidence values for each study, facilitating a comparative analysis between the *x* and *y* axes (see Appendix A for the regression statistics of the nationality, cad method, year of publication, type of skin cancer and band region respectively) [156]. Deek’s funnel plot is a graphical representation of the relationship between the mean effect size and the standard error (SE). This figure presents a comparison of the degree of variance seen across several studies [157]. The funnel plot displays symmetrical findings, indicating an equal distribution of studies above and below the mean regression line, which represents the standard error versus the odds ratio (please refer to Appendix A for the anova analysis of the nationality, cad methods, year of publication, type of skin cancer and wavelength band region respectively). The analysis of the funnel plot suggests that both the SKL and CNN exhibit significant standard error values but very low odds ratios. SVM and DI models exhibit high standard error values and odd ratios that are somewhat less diverse, as shown by a *p*-value of 0.813591, which is above the threshold of 0.005. Consequently, these findings suggest a higher level of heterogeneity, with a standard error of 4.39553. In comparison, Western research has higher standard error values and odds ratios when compared to Asian studies, with a regression line of 1.025.

While the systematic literature review and meta-analysis have provided valuable insights into the current landscape of skin cancer detection using HSI, several avenues for future research emerge. First, efforts should be directed toward standardizing data collection protocols and imaging technologies, as the heterogeneity in studies can impact the comparability of results. Second, exploring the integration of AI and machine learning algorithms with HSI for real-time and automated skin cancer detection holds significant promise. Third, enhancing the portability and affordability of HSI devices can expand their accessibility in clinical settings. Additionally, the development of user-friendly software tools for HSI data analysis and interpretation is essential. Further studies could also investigate the use of HSI in diverse skin types and ethnic populations to assess its robustness across demographics. Lastly, collaborative, interdisciplinary research involving clinicians, engineers, and data scientists is crucial for advancing the field and translating HSI-based skin cancer detection into routine clinical practice. The proposed approach in this study is not without its limitations, which are essential to acknowledge for a comprehensive understanding of the research. While the systematic literature review and meta-analysis have provided valuable insights, it is important to note that the quality and heterogeneity of the included studies can influence the generalizability of the findings. The variability in data collection protocols, imaging technologies, and sample sizes across the selected studies may introduce some degree of bias and limit the direct comparability of results. Additionally, the evolving nature of HSI technology and the relatively limited number of studies available for analysis may have implications for the current state of the field. Furthermore, the majority of the studies reviewed predominantly focused on specific skin cancer types or populations, potentially affecting the applicability of our conclusions to broader contexts. Lastly, while promising trends and potential areas of improvement have been identified, the translation of HSI-based skin cancer detection into clinical practice may encounter practical challenges, such as cost-effectiveness and regulatory considerations. This scarcity of eligible studies highlights the nascent stage of HSI research in this specific application. Acknowledging these limitations is crucial for a nuanced interpretation of our findings and underscores the need for further research to address these challenges and refine the use of HSI in clinical applications.

## 6. Conclusions

In conclusion, recent advancements in HSI for the identification of skin cancer have showcased promising outcomes, offering hope for early diagnosis and individualized treatment approaches. The progression in this field has been evident through the continuous improvement in research outcomes over time. The integration of CAD algorithms has the potential to significantly enhance the accuracy of skin cancer diagnosis. In this study, we have systematically categorized and examined various aspects of research in the domain of HSI for skin cancer detection, including the types of AI models used, the nationalities of researchers, publication years, and geographical regions of the studies. This comprehensive analysis serves as a testament to the global efforts and diverse approaches undertaken to harness HSI for skin cancer detection. However, it is important to acknowledge the existing limitations in this field. Constraints such as limited patient involvement and the scarcity of accessible imaging data and training datasets have been observed in relevant research. These challenges highlight the need for further collaboration, data sharing, and patient engagement to optimize the utility of HSI technologies in skin cancer detection. As we move forward, continued research and innovation in HSI, combined with a commitment to addressing these constraints, will be instrumental in advancing the early diagnosis and personalized treatment of skin cancer. HSI studies have demonstrated improved accuracy in the early detection and differentiation of skin cancer types, such as melanoma, basal cell carcinoma, and squamous cell carcinoma. The use of spectral data allows for more precise characterization of lesions, leading to fewer false positives and false negatives in diagnosis. HSI offers a non-invasive imaging technique, which means that patients do not have to undergo painful or invasive procedures for diagnosis. This is particularly important for skin cancer, where early detection can prevent the need for more aggressive treatments. The information obtained through HSI can aid in the development of customized treatment plans for individual patients. With a collective effort from researchers, healthcare professionals, and technology developers, we can look forward to a future where HSI plays a pivotal role in saving lives through early detection and improved patient care in the fight against skin cancer.

## Figures and Tables

**Figure 1 cancers-15-05634-f001:**
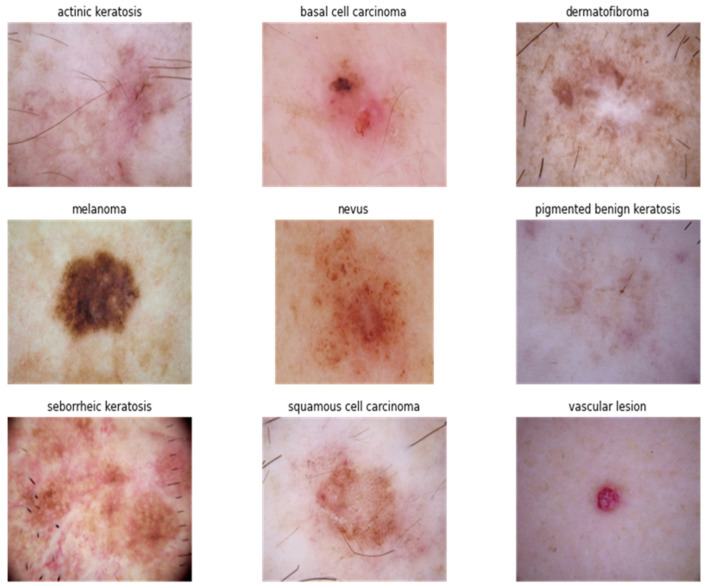
Types of skin cancers retrieved from the ISIC dataset.

**Figure 2 cancers-15-05634-f002:**
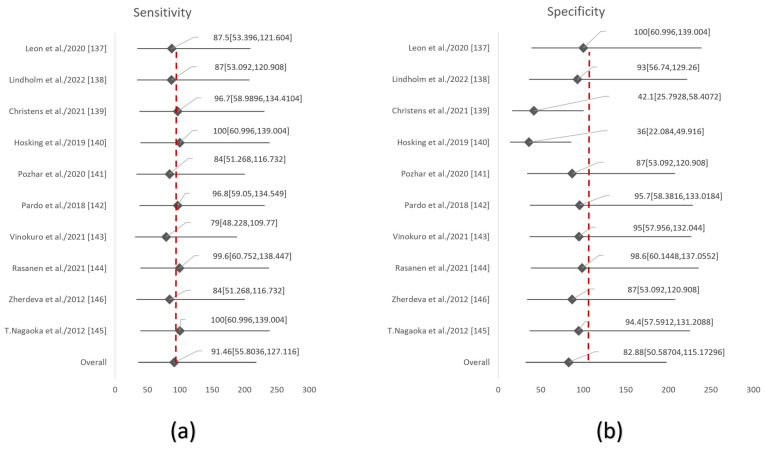
Forest plots based on the (**a**) sensitivities and (**b**) specificities of overall studies.

**Figure 3 cancers-15-05634-f003:**
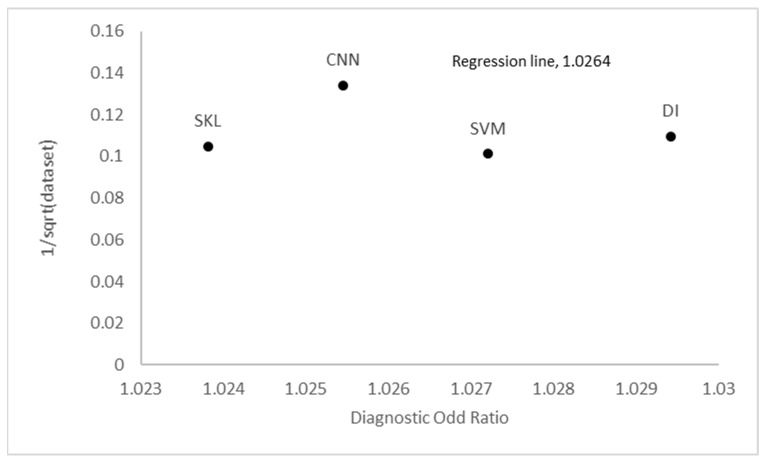
Deek’s funnel plot based on CAD methods.

**Table 1 cancers-15-05634-t001:** QUADAS-2 Summary.

	Risk of Bias	Applicability Concerns
Study	Patient Selection	IndexTest	ReferenceStandard	Flow andTiming	PatientSelection	IndexTest	ReferenceStandard
Leon et al. [136]	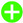	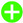	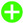	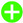	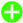	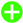	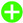
Lindholm et al. [137]	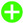	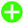	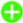	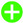	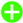	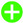	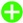
Christensen et al. [138]	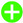	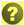	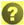	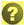	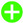	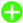	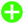
Hosking et al. [139]	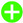	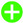	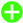	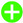	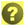	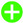	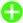
Pozhar et al. [140]	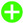	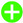	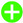	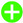	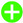	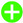	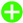
Pardo et al. [141]	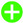	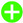	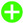	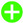	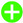	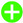	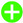
Vinokurov et al. [142]	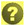	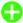	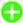	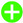	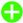	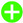	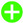
Rasanen et al. [143]	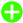	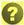	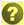	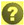	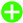	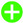	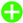
T. Nagaoka et al. [144]	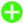	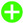	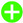	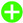	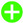	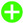	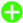
Zherdeva et al. [145]	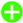	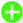	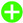	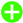	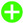	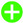	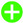

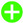
 Low Risk; 
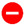
 High Risk; 
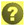
 Unclear Risk.

**Table 2 cancers-15-05634-t002:** Studies under clinical feature observation.

Author/ Year	No. of Patients	No. of Images	Wavelength (nm)	Sensitivity (%)	Specificity (%)	Nationality of Data	Type of AI	Year of Publication	Area under Curve (AUC)
Leon et al./2020 [136]	61	76	450–950	87.5	100	Western	SVM	2020	0.89
Lindholm et al./2022 [137]	33	42	477–891	87	93	Western	CNN	2022	NA
Christens et al./2021 [138]	186	202	400–800	96.7	42.1	Asian	DI	2021	0.800
Hosking et al./2019 [139]	91	100	350–950	100	36	Western	SVM	2019	1
Pozhar et al./2020 [140]	91	91	450–750	84	87	Western	SKL	2020	NA
Pardo et al./2018 [141]	116	124	600–900	96.8	95.7	Asian	CNN	2018	NA
Vinokuro et al./2021 [142]	NA	648	450–950	79	95	Asian	k-NN	2021	NA
Rasanen et al./2021 [143]	24	26	500–850	99.6	98.6	Western	CNN	2021	0.95
Zherdeva et al./2012 [145]	NA	45	450–750	84	87	Asian	DI	2012	NA
T. Nagaoka et al./2012 [144]	27	27	450–1000	100	94.4	Asian	NA	2012	NA

**Table 3 cancers-15-05634-t003:** Subgroup and diagnostic test accuracy meta-analysis.

Subgroup	Number of Studies	Number of Patients	Number of Images	Sensitivity (%)	Specificity (%)
Average meta-analysis of all studies	10	86	~150	91.89	84.28
Nationality of Data					
Asian	5	151	255	92.12	85.42
Western	5	50	67	76.35	82.92
Type of AI					
CNN	3	74.5	83	91.9	94.35
SVM	2	76	88	94.56	78.1
SKL	1	91	91	84	87
DI	2	186	123	90.35	64.55
Year of Publication					
2021–2022	4	81	~229	90.57	82.17
2019–2020	3	81	89	90.5	74.33
Before 2018	3	116	~84	94.25	93.85
Type of Cancer Classification					
BCC	3	81	90	94.43	77.9
SCC	2	NA	45	90.1	92.65
AK	1	91	91	84	87
Melanoma	4	104	113	98.62	73.36
Band Region					
Visible (380–780 nm)	10	86.5	175	80.4	92.66
Visible near IR	8	92.75	105	94.567	77.3

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
