# Peer review of "A Review of Recent Advances in Computer-Aided Detection Methods Using Hyperspectral Imaging Engineering to Detect Skin Cancer"

_cancers, 2023, doi:10.3390/cancers15235634_

Round 1

Reviewer 1 Report

Comments and Suggestions for Authors

This study reports on recent CAD studies related to skin cancer detection and diagnosis using HSI based on their diagnostic performances in terms of sensitivity and specificity. However, the work does not contribute much to computer-aided detection methods in any meaningful way. In terms of skin cancer, the review does not adhere to the standards of solid scientific practice. The conclusion section provides general information. The work concentrates almost entirely on sensitivity and specificity; it does not take into account any other metrics. The contents of the dataset as well as the criteria for research selection are not thoroughly and properly disclosed. It is necessary to update the sections on the methodology and results with the most recent studies.

The abstract should be improved.

The methodology and results section should be revised or rewritten completely.

The discussion and conclusion sections should be improved. The conclusion should focus on the proposed review approach and the achievements of current and previous work.

Comments on the Quality of English Language

Minor editing of English language required

Reviewer 2 Report

Comments and Suggestions for Authors

The detailed comments on the proposed approach are enlisted below. Although the authors have written a sound and thorough approach, some comments are presented below to improve the quality of the manuscript.

REVIEWER REPORT

Article Title: A Review of Recent Advances in Computer-Aided Detection 2 Methods using Hyperspectral Imaging Engineering to Detect 3 Skin Cancer

• The manuscript lacks a clear motivation, which motivates the authors to propose this approach.

• The key contributions should be highlighted in the introduction, currently, the manuscript misses it.

• The introduction section of the paper is quite long; it should be divided into two sections, i.e., Introduction and Background.

• The structure of the manuscript is missing at the end of the introduction.

• A detailed research approach describing all the steps should be drawn and put in the paper to have a better understanding of the proposed research approach to the readers.

• Did the authors use a snowballing approach for selecting the papers? The authors didn’t mention the common and signature journals and conferences where they can find up-todate papers on the research topic.

• The authors didn’t show the year-wise distribution of the papers.

• How the quality of the papers selected for the research approach is measured?

• The manuscript does not have a related work section, which is an important part of the paper, where the author elaborates on the existing approaches and highlights the proposed approach's advantages over the existing approaches.

• At the end of the related work section, there should be a paragraph explaining the proposed method compared to the state-of-the-art and why your approach is different from those explained and compared in the related work section.

• An important part of a review paper is highlighting future directions, which helps researchers focus on it and propose possible solutions for the problems mentioned and identified through SLR.

• The proposed approach lacks the limitations of the proposed study. Clarifying the study's limitations allows the readers to understand better under which conditions the results should be interpreted. A clear description of the limitations of a study also shows that the researcher has a holistic understanding of his/her study. However, the authors fail to demonstrate this in their paper.

• The research study is limited to very few analyses and results. It should be extended in various direction to benefit potential readers to get more insightful information from the article on the field of discussion.

• It is recommended to consult several more recent articles in this field as an example when Rearranging and upgrading the related work section of this article. For instance, you can Use as an example the below article: DeepBreastCancerNet: A Novel Deep Learning Model for Breast Cancer Detection Using Ultrasound Images

Deep learning enables automated MRI-based estimation of uterine volume also in patients with uterine fibroids undergoing high-intensity focused ultrasound therapy. A Robust End-to-End Deep Learning-Based Approach for Effective and Reliable BTD Using MR Images

Reviewer 3 Report

Comments and Suggestions for Authors

- Why only sensitivity and specificity performance metrics are selected? Justification and discussion on this selection must be provided and supported by relevant references.

- The layout of the study could be explained at the end of the introduction section.

- It is stated under Section 2.1 that "the studies should have been published within the last five years" but there are two studies from 2012 discussed in this manuscript.

- What is the justification for the study selection criteria? especially where it is stated, "the selected journals should have an impact factor exceeding 3, an H-index greater than 50, and fall within the first quartile (Q1) of their respective fields." Justification is needed. Many good studies can be found in other quartiles, especially in Q2.

- There are several studies related to skin cancer detection using Hyperspectral Imaging 2023. But none of them are considered in the comparison.

Round 2

Reviewer 1 Report

Comments and Suggestions for Authors

This comment still has not improved: The methodology and results section should be revised or rewritten completely. In response, the authors only rewrote Section 2.1, Study Selection Criteria.

Reviewer 2 Report

Comments and Suggestions for Authors

The authors have address some of the comments, while few comments have been left un-answered that needs to be incorporated in the review. It is advised to take all the previous comments in consideration. Currently, some comments are unaddressed or the response letter does not compliance to the version uploaded. Please provide with the updated and correct version if changes are incorporated in the paper. 

Reviewer 3 Report

Comments and Suggestions for Authors

- Requested justifications for performance metrics and study selection criteria should be added to the paper not only added to the response letter.

- Justification for not adding studies from 2023 is not acceptable. This causes the study to not be up-to-date. Many of the published studies in 2023, in fact, they were conducted much earlier and maybe in 2022. This is the same for the studies published in 2022 and earlier. 

Round 3

Reviewer 2 Report

Comments and Suggestions for Authors

Although the authors improved the quality of the paper, some of my previous comments haven't been incorporated into the paper. It is, therefore, recommended to consider all my previous comments.

Reviewer 3 Report

Comments and Suggestions for Authors

Related studies published in 2023 should be included. A simple search using the study's keywords shows at least a few relevant studies that could be reviewed and included in this manuscript.
